# Microbial Co-Occurrence Patterns and Keystone Species in the Gut Microbial Community of Mice in Response to Stress and Chondroitin Sulfate Disaccharide

**DOI:** 10.3390/ijms20092130

**Published:** 2019-04-30

**Authors:** Fang Liu, Zhaojie Li, Xiong Wang, Changhu Xue, Qingjuan Tang, Robert W. Li

**Affiliations:** 1College of Food Science and Engineering, Ocean University of China, Qingdao 266003, China; liufang910205@163.com (F.L.); lizhaojie@ouc.edu.cn (Z.L.); wangxiong9202@163.com (X.W.); chxue@ouc.edu.cn (C.X.); 2United States Department of Agriculture, Agriculture Research Service (USDA-ARS), Animal Genomics and Improvement Laboratory, Beltsville, MD 20705, USA

**Keywords:** 16S rRNA gene, chondroitin sulfate disaccharide, co-occurrence network, global network, microbial interactions, microbiome, modularity, superoxide dismutase

## Abstract

Detecting microbial interactions is essential to the understanding of the structure and function of the gut microbiome. In this study, microbial co-occurrence patterns were inferred using a random matrix theory based approach in the gut microbiome of mice in response to chondroitin sulfate disaccharide (CSD) under healthy and stressed conditions. The exercise stress disrupted the network composition and microbial co-occurrence patterns. Thirty-four Operational Taxonomic Units (OTU) were identified as module hubs and connectors, likely acting as generalists in the microbial community. *Mucispirillum*
*schaedleri* acted as a connector in the stressed network in response to CSD supplement and may play a key role in bridging intimate interactions between the host and its microbiome. Several modules correlated with physiological parameters were detected. For example, Modules M02 (under stress) and S05 (stress + CSD) were strongly correlated with blood urea nitrogen levels (*r* = 0.90 and −0.75, respectively). A positive correlation between node connectivity of the OTUs assigned to Proteobacteria with superoxide dismutase activities under stress (*r* = 0.57, *p* < 0.05) provided further evidence that Proteobacteria can be developed as a potential pathological marker. Our findings provided novel insights into gut microbial interactions and may facilitate future endeavor in microbial community engineering.

## 1. Introduction

Glucosamine (GS) and chondroitin sulfate (CS), a large class of sulfated glycosaminoglycans, are major structural components of joint cartilage and have been widely used as a dietary supplement for maintaining cartilage integrity as well as alleviating osteoarthritis symptoms. CS possesses various biological functions by acting as extracellular signaling molecules/modulators and co-receptors, in addition to their structural role [1]. CS and its component oligosaccharides have different modulatory effects on the structure and function of the gut microbiome [2,3,4,5]. Chondroitin sulfate component disaccharide (CSD) can significantly modify the function of gut microbiome and increase intestinal *Bacteroides acidifaciens* populations in a rodent model [2]. Moreover, our previous results show that CSD has the potential to change kidney morphology and repair kidney cortex damaged by exhaustive exercise stress. CSD dietary treatments significantly decrease blood urea nitrogen (BUN) levels (*p* < 0.05) [2]. A randomized human trial demonstrated that glucosamine and chondroitin supplementation lowers systemic inflammation and reduces oxidative stress, as measured by urinary prostaglandin F2α [6]. Malondialdehydes (MDA) are one of the frequently used markers of oxidative stress in response to exercise [7]. Free radicals generated during the stress attack polyunsaturated fatty acids in the cell membrane, leading to a chain of chemical reactions termed lipid peroxidation. Superoxide dismutase (SOD) acts as one of the key enzymatic antioxidant defenses against superoxide radicals [8]. Endurance exercise generally results in an increase in total activities of superoxide dismutase (SOD), which acts as one of the key enzymatic antioxidant defenses against superoxide radicals [8] and leads to enhanced resistance to oxidative stress [9]. In pigs, long-term aerobic exercise increases SOD1 protein concentrations as well as SOD1 enzyme activities, which tends to lower biomolecular indexes of oxidative stress, as reflected by decreased MDA values [10].

While effectiveness of CSD and GS supplements is still debated, several studies have raised questions on their potential renal toxicity [11]. Renal dysfunction and cardiovascular diseases are two of the serious medical concerns. Blood urea nitrogen (BUN) is an important physiological parameter of health; and long-term elevated BUN levels are associated with an increased risk of cardiovascular and renal conditions [12,13]. For example, BUN acts as a significant prognostic factor for the mortality in patients with acute ischemic stroke [14]. Moreover, BUN levels, especially the BUN to creatinine ratio (CR), are correlated with renal health. Recent studies suggest that the intestinal microbiome play critical roles in maintaining renal function [15,16]. The rats treated with *Lactobacillus* probiotics displayed improved BUN values and may ameliorate renal damage [17]. Gut microbiome-derived metabolites, such as trimethylamine-N-oxide and short-chain fatty acids, also play an important role in cardiovascular diseases [18]. As a result, targeting gut microbiota has become a potentially promising therapy for diabetic kidney disease recently [19].

The detection of species interactions and interdependence in a microbial community is beyond the scope of traditional alpha and beta diversity metrics. Network analysis tools enable a better understanding of microbial interactions and potential ecological roles of keystone species in complex microbial communities [20]. However, little is known about biotic interactions among different microbial species in the gut microbial communities in response to CSD and/or GS. Moreover, co-occurrence patterns between various microbial taxa in these microbial communities have not been explored; and microbial taxa that may be significantly correlated with various physiological parameters, such as BUN, SOD, and MDA, have not been identified. In this study, we attempted to understand microbial interactions and identify key microbial taxa that may be strongly correlated with BUN as well as other physiological parameters, such as SOD and MDA, in response to CSD dietary intervention under both healthy and exhaustive exercise-induced stressed conditions using co-occurrence network tools.

## 2. Results

### 2.1. Stress from Exhaustive Exercise Induced a Distinctly Different Microbial Co-Occurrence Network in Mice

The mean number of raw reads (2 × 300 bp paired-end) for the dataset is 227,448 ± 98,540 per sample (*n* = 30). After various quality control procedures, including trimming and merging of the pair-ended reads, rarefaction was performed at a depth of 100,000 retained quality sequences. Both closed and open-reference protocols in the QIIME pipeline were used for OTU picking at 97% similarity. A total of 2257 OTU was obtained for the study (Appendix A). The numbers of OTU in the input datasets used for global network inference were 915 (Group N), 925 (Group M), 854 (Group C), and 844 (Group S), respectively. Global microbial co-occurrence networks were constructed using a Random Matrix Theory (RMT)-based method, as described in the Phylogenetic Molecular Ecological Network pipeline [21,22]. The mean numbers of nodes per module ranged from 7.80 (the S group) to 12.37 in the C group. Topological properties of microbial co-occurrence networks inferred in the four experimental groups were described in Appendix A. The numbers of nodes in the global networks inferred from the normal healthy groups (N) and exercise-induced stress group (M) were 539 (788) and 502 (782), respectively. Among them, 340 OTUs (nodes), accounting for 63% and 68% of all nodes in global networks in N and M groups, respectively, were shared by both networks. While the overall network sizes between the two groups appeared to be similar, the network composition and structure were distinctly different. Individual modules differed in the number of nodes (size) and shape (connections) in both groups. Approximately half of all modules, 27 in total, in each network contained only 2 to 3 members. These small modules were generally isolated with no links to the remaining network. There were 20 and 21 modules with six or more members in N and M groups, respectively. The node composition in global networks of each experimental group was substantially different. Many modules in each network was unique. For example, 16 of the 26 members consisting of the module N7 were unique to the N network, while eight of the 13 members in the module N14 was unique. Similarly, six of the seven members in the module M22 were unique to its own network. Very few compositional (or functional) equivalent modules can be paired in the two global networks. One exception may be N2 and M1 modules in N and M global networks. Among 30 and 38 members in N2 and M1 modules, respectively, 21 members were shared between the two networks, suggesting that the two modules may be functional equivalents. The phylogenetic composition of the nodes in each global network appeared relatively stable. For example, the most predominant phylum in each of the four global networks, *Firmicutes*, accounted for approximately 75.5% of all nodes in the networks. However, the percentage of *Proteobacteria* in each of the global networks varied substantially. The stress appeared to increase the percentage of *Proteobacteria* from 5.2% in normal/healthy group (N) to 6.77% in the stressed network (M). Intriguingly, CSD dietary treatments decreased the percentage of nodes assigned to the phylum *Proteobacteria*, from 5.2% to 3.9% under the healthy condition, and 6.8% to 4.1% under the stressed condition.

### 2.2. Keystone Species and their Possible Ecological Roles

Nodes play different topological roles in the network. Plots generated based on within-module degree z and among-module connectivity p, allow us to identify some key nodes in the network. As shown in Appendix A, the majority of the nodes, accounting for >98% of all nodes, in the global networks generated from the 4 experimental groups were peripherals with low *Zi* and low *Pi* values. Among them, between 79% to 92% of all nodes had no links with outside of their own modules (*Pi* = 0). Between seven and 11 nodes were identified that may serve as module hubs in the four networks. These nodes represented approximately 1.3% to 2.2% of all nodes in the networks, in a good agreement with the findings in other environmental microbial communities [22]. These nodes were highly connected and tended to link to other nodes within their own module (a high *Zi* > 2.5 and a low *Pi* ≤ 0.62) which may be important to the coherence of their own modules. For example, an OTU from the genus *Bacteroides* (GreenGeneID# 1135084) and OTU# (GreengeneID#183321) from *Prevotella*, were module hubs in the network from the C group while an OTU from *Ruminococcus* (GreenGeneID# 339031) and an OTU from *Lactococcus* (#586387) behaved like module hubs only in the healthy group (N). The relative abundance of the majority of the module hubs, six out of the eight OTUs, identified in the C group were significantly changed by CSD (absolute log_10_ LDA score > 2.0). For example, the abundance of OTU# 1135084 was significantly increased by CSD under both healthy and stressed conditions [2]. Intriguingly, the module hubs were distinctly different in each of the four global networks. Only one node, an OTU (GreenGeneID# 766563), belonging to the genus *Leuconostoc*, acted as a module hub in the networks from both healthy and stressed groups. These findings suggested that the hubs of different global networks and resultant network structure under different stress and dietary conditions were substantially different. Only a few nodes acting as a connector that linked different modules together was identified in scatter plots (Figure 1). Of note, an OTU, GreenGeneID#549991, belonging to the genus *Lactobacillus* acted as a connector species, which links two or more different modules together, under the healthy condition, was identified. The abundance of this OTU was significantly depressed by the exercise stress (LDA score > 2.0). Another OTU (GreenGeneID#1136443), assigned to *Mucispirillum schaedleri*, served as a connector in the global network from the S group. A network hub, the node with a very high between-module and among-module connectivity value, is likely important to the coherence of the global network and as well as its own module and can be considered keystone species in a microbial community [23]. However, no network hubs were identified in each of the four global networks analyzed in this study.

### 2.3. The Correlations between Modules and Physiological Parameters

To understand the response of individual modules to physiological parameters, the correlations between module-based eigengenes and the four physiological measurements were calculated (Table 1). Under normal or healthy condition (N), at least 4 modules were positively correlated with three physiological traits. Both the modules N01 and N05 (Figure 2) in the healthy network and Modules C15 and S05 of the groups C and S (Figure 3), respectively, were strongly correlated with the BUN value (*p* < 0.05), while the module N14 was correlated with CR. Several modules in the group M network (Figure 4) also displayed a strong correlation with either BUN (Module M03) or creatinine ratio (CR, Module M13) alone or both (Modules M02 and M21). At least two modules, N09 and M27, had a strong correlation with the kidney concentration of malondialdehyde (MDA), an important lipid peroxidation marker (Figure 5). In the experimental group C, the module C10 (Figure 6) showed a strong negative correlation with SOD (*r* = −0.94; *p* = 0.006), while module C15 was positively correlated with BUN (*r* = 0.89; *p* = 0.02). Among the eight members consisting of the module C10, the OTUs from the phylum *Bacteroidetes* were predominant; and six of the eight members belonged to the order *Bacteroidales*, including at least 2 OTUs assigned to *Bacteroides acidifaciens*. Moreover, the relative abundance of at least 5 OTUs belonging to *Bacteroidetes* were significantly affected by the CSD treatment. In the stressed group (M), the module M02 displayed a strong (*r* = 0.90) and significant positive correlation with both BUN and CR values (*p* = 0.002) (Table 1). Similarly, the module M21 (Figure 4) was positively correlated with both BUN and CR (*r* = 0.87; *p* = 0.005). The module M03 was negatively correlated with the BUN value (*r* = −0.89; *p* = 0.003). Moreover, the module M01 were positively correlated with the SOD value (*r* = 0.71; *p* = 0.05) while M14 was negatively correlated with SOD (Figure 6). The module M01 contained a total of 38 members, including eight members from the phylum *Bacteroidetes*, 27 members from *Firmicutes*; and two members from *Proteobacteria*. All eight members of the phylum *Bacteroidetes* belong to the family *S24-7*. In the experimental group S, the module S05 was negatively correlated with BUN (*r* = −0.75; *p* = 0.05).

The relationship of network topology and physiological parameters were also assessed by calculating the correlation between the OTU significance (*GS*), *r*^2^ (the square of Pearson correlation coefficients) of OTU abundance profiles with traits and node connectivity. Under the normal or healthy condition, the node connectivity of the OTUs assigned to the order *Pseudomonodales* as well as that of the OTUs assigned to the families *Porphyromonadaceae* and *Moraxellacease* was significantly correlated with *GS* of the BUN value (Table 2). Under the stressed condition (M), the node connectivity of the phylum *Proteobacteria* was positively correlated with changes in the SOD value (*p* < 0.05). With CSD treatment under both normal and stressed conditions (C and S), the node connectivity of the OTUs belonging to *Pseudomonodales* became positively correlated with CR values, similar to what happened under the normal condition (N). Moreover, the node connectivity of the OTUs in the family *Lactobacillacease* was positively but marginally correlated with both BUN and SOD values (*p* < 0.05 in both cases).

## 3. Discussion

Over the past few years, various computational algorithms have been developed to infer microbial co-occurrence networks from the microbiome data [24,25,26,27]. While different methods, no matter how they measure various features, either Bray-Curtis abundance similarity, Pearson or Spearman correlation coefficients, or Maximal Information coefficients, varied substantially in sensitivity and precision [28], they had common goals to detect robust microbial associations and infer co-occurrence network patterns between microbial entities in a given habitat or environment. The knowledge obtained can provide novel insights into the organization of complex microbial communities and deciphering key microbial populations and their ecological roles. The network approach may allow us to understand functional roles of a closed related group of microbes, especially those unculturable or elusive but ecologically relevant species. In addition, characterization of species interactions or interdependence in a gut microbial community will undoubtedly enrich our understanding of true microbial diversity. In this study, we used a Random matrix theory (RMT) based on method (MENA) to infer microbial co-occurrence networks in fecal microbiome samples of the mice in response to chondroitin sulfate disaccharide dietary treatments under both healthy and stressed conditions. One of the unique features of this RMT method is automatic detection of similarity threshold values by calculating the transition from Gaussian to Poisson distributions [21,22]. This method, while paired with Pearson correlations, significantly improves the precision and tends to generate the fewest false positives [28].

Our previous study demonstrates that exhaustive exercise stress has a profound impact on the structure and function of the murine gut microbiome, resulting in a significant alteration of the fecal microbial community, including a significant change in relative abundance of 4 genera (*Aeromicrobium*, *Anaerostipes*, and *Turicibacter,* and *Anaerotruncus*) and 76 OTUs [2]. Of note, the abundance of at least 10 OTUs in the family *S24-7* was significantly repressed by the stress. In this study, our results suggest that while the overall size and structure of the global networks appeared to be similar, the network composition was distinctly different between healthy and stressed conditions. The networks from each condition had a similar number of modules or subnetworks, especially the large modules with six or more members (20 and 21 larger modules for healthy and stressed conditions, respectively). However, only approximately 60% of all nodes were common components in both networks. Moreover, none of the modules appeared to be paired with similar node compositions, suggesting that the exercise stress significantly disrupted the network structure and composition. Furthermore, compared to the healthy condition, the percentage of the nodes (OTUs) assigned to *Proteobacteria* was much higher in the network from the stressed condition. The increased *Proteobacteria* abundance is a striking feature of an unstable microbial community and associated with inflammation [29,30]. Intriguingly, the percentage of the nodes belonging to *Proteobacteria* in the networks was significantly reduced by the CSD intervention under both conditions. Our previous study identified a significant reduction in the relative abundance of *Proteobacteria* by CSD [2], which may contribute to its anti-inflammatory properties. Together, our findings suggest that CSD supplementation had a potential to reduce both relative abundance and microbial interactions of pro-inflammatory *Proteobacteria*.

Modularity, the extent to which species interaction are organized into modules or subnetworks in a network, may reflect habitat heterogeneity or phylogenetic clustering of closely related species; and modules with closed linked species may represent some key coevolution units [31]. In this study, all four global networks were identified with a very high modularity, ranging from 0.8 to 0.9. In these networks, greater than 98% of the OTUs were peripherals, likely acted as specialists in the microbial community. Collectively, 32 OTUs were identified as module hubs or highly connected species linked to other nodes in their own modules. Only two OTUs were connector species that linked several modules together. Module hubs and connectors can serve as generalists in the microbial community. An OTU belonging to *M. schaedleri* acted as a connector species in the S network. As a member of the core murine gut microbiome, this species is residing in the mucus layer of the gastrointestinal tract. The increased abundance of *M. schaedleri* has been documented in chemically reduced colitis and inflammatory models [32,33]. The species is known to express secretion systems and effector proteins, which can modify the gene expression of host mucosa. Moreover, this species may possess capacity to degrade mucin. Together, these data suggest that *M. schaedleri* undergoes intimate interactions with its host and may play a role in inflammation. Many of the generalists identified in this study acted as module hubs. Moreover, each network had a distinct set of module hubs, which likely reflected habitat heterogeneity or trophic specialization under different experimental conditions. The OTU# 1135084, assigned to the genus *Bacteroides*, which was significantly increased by CSD supplementation under both healthy and stressed conditions [2], acted as a module hub in the module C9 in the C group. Many *Bacteroides* species possess species-specific dynamics responses to CS or CSD availability [34]. For example, *B. thetaiotaomicron*, can rapidly activate the transcription of CS utilization genes after a sudden exposure to CS and then dynamically adjust their transcription in response to the rate at which CS is broken down [35]. *B. acidifaciens* is a predominant member colonized in the murine gut and possesses strong immunomodulating activities. The abundance of several OTU assigned to *B. acidifaciens* was significantly increased by the CSD supplement [2]. The effect of niche disturbances or perturbation likely spreads more slowly in a modular structure [31]. The elimination or a significant reduction of a module hub may cause a modular structure to collapse without a major cascading effect on other modules. Likewise, the expansion of a module hub, such as the one in the genus *Bacteroides*, can have a broad effect on microbial interactions, especially on the species of its own module. As a result, dietary interventions targeting on generalists or super generalists, such as connectors or key module hubs, may have a higher success rate to achieve a desired effect.

The correlation analysis between module-based eigengenes and physiological traits was used to aid the understanding of the responses of individual modules to changes in physiological parameters. For example, numerous published reports support the notion that glucosamine and chondroitin, alone or in combination, possess chondroprotective properties, including a significant reduction of osteoarthritis symptoms with less adverse events [36]. However, several studies suggest a possible link between these supplements and renal dysfunction, manifested in part by elevated levels of BUN and creatinine [37]. In this study, we detected several modules significantly correlated with BUN traits in each of the four networks. Under the healthy condition, Modules N01 and N05 were positively correlated with BUN (*p* < 0.05). The module M02 was strongly positively correlated with BUN (*r* = 0.90, *p* < 0.005) while the module M03 was strongly negatively correlated with BUN (*r* = −0.89, *p* < 0.005) under the stressed condition. Similarly, positive and negative correlations with BUN were detected in the eigengenes of the module C15 (*r* = 0.89, *p* < 0.05) and the module S5 (*r* = −0.75, *p* < 0.05) by the CSD treatment under healthy and stressed conditions, respectively. A close examination of the module membership failed to identify any shared module composition in these networks, even though they were all somehow correlated with BUN. Nevertheless, carefully designed experiments to gain a deep understanding of microbial interactions in these modules may unravel mechanisms of action of chondroitin supplements. In addition, positive and negative correlations were detected between the M01 and M14 eigengenes, respectively, and SOD values under the stressed condition. The Mantel test was used to calculate the correlations between the node connectivity and physiological parameters. Under the stressed condition, the node connectivity of the OTUs assigned to the phylum *Proteobacteria* was positively correlated with SOD (*r* = 0.57, *p* < 0.05). SOD is known to play an important role in inflammation and other diseases. Together, our findings provide further evidence that *Proteobacteria* can be developed as a stress or pathological marker. In addition, the findings suggest that altered network topologies, especially the node connectivity in some key modules, may have important implications on tissue and blood physiological parameters. It is well known that numerous biotic and abiotic factors affect the gut microbial composition and species interdependence in the gut microbial community. It is important to design focused experiments to validate the microbial interactions inferred using network tools. Nevertheless, the microbial co-occurrence network inference approach can provide us with novel insight into the potential functional role of key microbial taxa in not just microbial communities but also the complicated traits or physiological parameters of the host. The knowledge obtained via this approach should facilitate microbial community engineering by targeted elimination and expansion of keystone species in a network and will be of practical significance in modeling the effect of a successful dietary intervention.

## 4. Materials and Methods

### 4.1. Animals Experiment

The animal experiment was previously described [2]. Briefly, 30 Balb/c mice were housed in the same room and fed the same basal diet. The mice were randomly assigned to 4 groups: Healthy control mice + Phosphate-buffered saline or PBS (the group N; *n* = 9), Exhaustive exercise stressed mice + PBS (the group M; *n* = 8), Healthy mice supplemented with CSD at a daily dose of 150 mg/kg body weight for 16 consecutive days (the group C; *n* = 6); and Exhaustive exercise stressed mice supplemented with CSD at a daily dose of 150 mg/kg body weight for 16 days (the group S; *n* = 7). Animals in both M and S groups were subjected to a forced exercise wheel-track treadmill. After an initial resting period to collect baseline data, exercise commenced at 20 rpm running speed for 3 h each day for 2 days. A recovery and resting period of 5 days was allowed after the 2-day exercise. The entire experiment lasted for 16 days, including 6 days in which exhaustive exercise was endured. The CSD dietary supplement was initiated at the same time as the exercise stress commenced and lasted for 16 days. The animal protocol (Project# SCXK-Jing-2007-0001G) was approved by the Committee on the Ethics of Animal Experiments of Ocean University of China (approval date: 27 December 2007); and the experiment was conducted by strictly following the Institutional Animal Care and Use Committee guideline (IACUC).

### 4.2. Physiological Parameters

All assay kits were purchased from Nanjing Jiancheng Bioengineering Institute (NJBI, Nanjing, China). Blood creatinine kinase (CK) activities and BUN values were measured as previously reported [2]. All physiological parameter data can be found in the Appendix A.

SOD activities and MDA concentrations in the renal tissue were measured using assay kits purchased from NJBI. Total SOD activities were measured based on a WST1 [2-(4-iodophenyl)-3-(4-nitrophenyl)-5-(2,4-disulfo-phenyl)-2H-tetrazolium, monosodium salt] SOD inhibition assay using a colorimetric microplate reader. Briefly, renal tissue samples were weighed and homogenized in a microfuge tube immersed in an ice slurry. The homogenate was then incubated at 4 °C for 30 min. After a 5-min centrifugation at 12,000× g (4 °C) to pellet tissue debris, the supernatant was transferred to a new tube for assay analysis. The SOD activity was defined as units per mg protein.

### 4.3. 16S rRNA Gene Sequencing and Data Analysis

Feces were collected at necropsy and stored at −80 °C until total DNA was extracted. Total DNA was extracted using a bead-beating method as described [2,38]. The quality of the total DNA was verified using a BioAnalyzer 2100 (Agilent, Palo Alto, CA, USA). DNA concentration was first measured using a Nanodrop instrument and then verified by a Quantus fluorometer (Promega, Madison, WI, USA). The hypervariable V1-V3 regions of the 16S rRNA gene were directly amplified from 20 ng of fecal total DNA with Polyacrylamide Gel Electrophoresis (PAGE)-purified Illumina platform-compatible adaptors that contain features such as sequencing primers, sample-specific barcodes, and 16S PCR primers (forward primer, 9F, GAGTTTGATCMTGGCTCAG; reverse primer, 515R: CCGCGGCKGCTGGCAC). The PCR reaction included 2.5 units of AccuPrime Taq DNA Polymerase High Fidelity (Invitrogen, Carlsbad, CA, USA) in a 50 μL reaction buffer containing 200 nM primers, 200 nM dNTP, 60 mM Tris-SO_4_, 18 mM (NH4)_2_SO_4_, 2.0 mM MgSO_4_, 1% glycerol, and 100 ng/uL bovine serum albumin (New England BioLabs, Ipswich, MA, USA). PCR was performed using the following cycling profile: initial denaturing at 95 °C for 2 min followed by 20 cycles of 95 °C 30 s, 60 °C 30 s, and 72 °C 60 s. Amplicons were purified using Agencourt AMPure XP beads (Beckman Coulter Genomics, Danvers, MA, USA), quantified using a BioAnalyzer high-sensitivity DNA chip, and pooled at equal molar ratios. The pool was then sequenced using an Illumina MiSeq sequencer, as described previously [38]. The raw sequences have been deposited to the NCBI Sequence Read Archive (*SRA*) database and are freely accessible (SRA accession# SRP137092)

### 4.4. Network Construction and Visualization

The quality control, preprocessing, and OTU picking steps were conducted using the Quantitative Insights Into Microbial Ecology (QIIME) pipeline (v1.9.1) [39]. Both “closed reference” and “open-reference” protocols in the pipeline were used for OTU picking. The global networks were constructed for each of the 4 experimental groups using a Random-Matrix theory (RMT) based pipeline described [21,22] (http://ieg4.rccc.ou.edu/mena/). The input datasets used for the network construction were the OTU abundance table derived using the “closed reference” protocol in the QIIME pipeline. Because the OTU sparsity has a drastic effect on the precision and sensitivity of network inference [28], the rare OTU, that is, those OTU detected in <50% of all samples, were excluded. While this practice may have a negative effect on network structure, it improves the false positive rate of interaction detection. A similarity matrix, which measures the degree of concordance between the abundance profiles of individual OTUs across different samples [21], was then obtained by using Pearson correlation analysis of the abundance data. A threshold was automatically determined by calculating the transition from Gaussian orthogonal ensemble to Poisson distribution of the nearest-neighbor spacing distribution of eigenvalues, in the pipeline and then applied to generate an adjacent matrix for network inference [22]. In this study, the stringent threshold with significance >0.05 was selected to control the false positive rate. The threshold values used in this study were 0.87 for the N group, 0.92 for M, 0.98 for C, and 0.94 for S, respectively. The fast-greedy modularity optimization procedure was used for module separation. The within-module degree (*Zi*) and among-module connectivity (*Pi*) were then calculated and plotted to generate a scatter plot for each network to gain insights into topological roles of individual nodes in the network. The Olesen classification approach was used to define node topological roles [31]. A partial Mantel test was performed to measure the relationship of the network topology and physiological traits by calculating OTU significance and node connectivity, as described [21]. Finally, the networks were visualized using Cytoscape v3.6.1 [40].

## 5. Conclusions

Global network analysis allows a better understanding of microbial interactions and potential ecological roles of keystone species in complex microbial communities in the gut. Under healthy condition, CSD alters the abundance of the majority of module hubs or connector species, such as those from *Bacteroides* and *Prevotella* while *Mucispirillum schaedleri*, a mucin-utilizing bacterium, acts as a connector in the network stressed by exercise. One of mechanisms by which CSD exerts its potential therapeutic effects is via modulation of network composition and microbial co-occurrence patterns significantly disrupted by exercise stress. The study highlights the importance of microbial community engineering based approaches in expanding the therapeutic potential of CSD to diseases and pathophysiological conditions with abnormal blood urea nitrogen levels and superoxide dismutase activities.

## Figures and Tables

**Figure 1 ijms-20-02130-f001:**
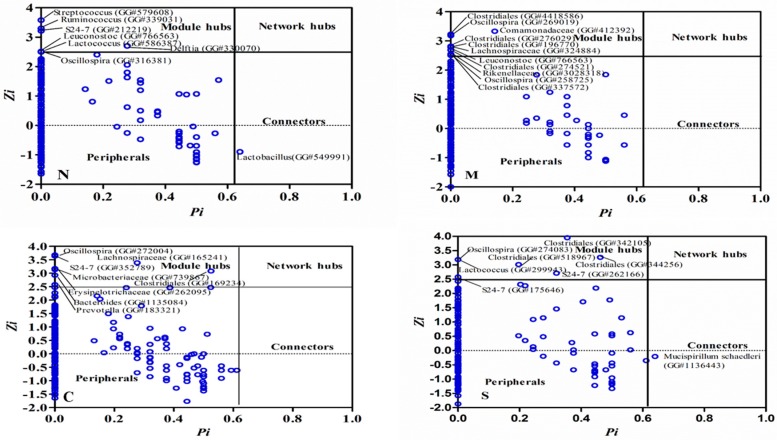
The scatter plot showing the distribution of OTU based on their topological roles in the network. The detailed information for each OTU can be found in Appendix A. Each dot represents an OTU. Zi: within-module connectivity. Pi: Among-module connectivity. The network topological role classification was originally proposed by Olesen et al., 2007. N: healthy mice supplemented with phosphate-buffered saline (PBS); C: healthy mice supplemented with a daily dose of 150 mg/kg bodyweight of chondroitin sulfate disaccharide (CSD) for 16 consecutive days; M: mice subjected to exhaustive exercise stress supplemented with PBS; S: the stressed mice supplemented with a daily dose of 150 mg/kg bodyweight of CSD for 16 days.

**Figure 2 ijms-20-02130-f002:**
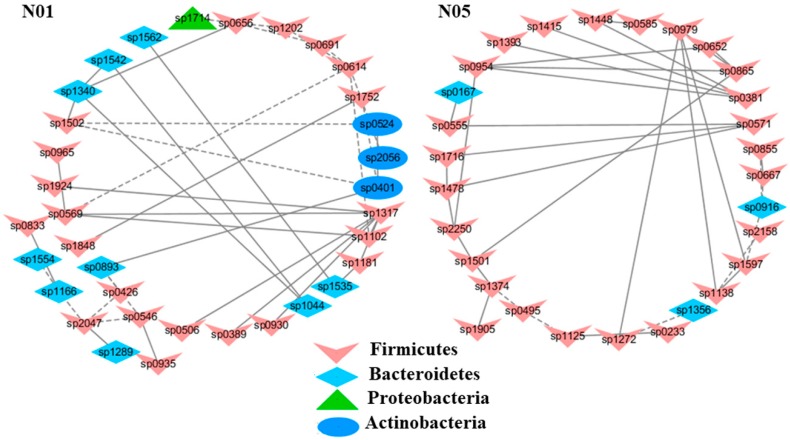
Select modules displaying a strong correlation with blood urea nitrogen contents (BUN) in healthy mice. The interactions among different nodes (OTUs) within a module were shown: solid line: positive correlation; dashed line: negative correlation. The color of each node (OTU) indicated the phylum that this OTU was assigned to. The detailed annotation of each OTU node can be found in Appendix A.

**Figure 3 ijms-20-02130-f003:**
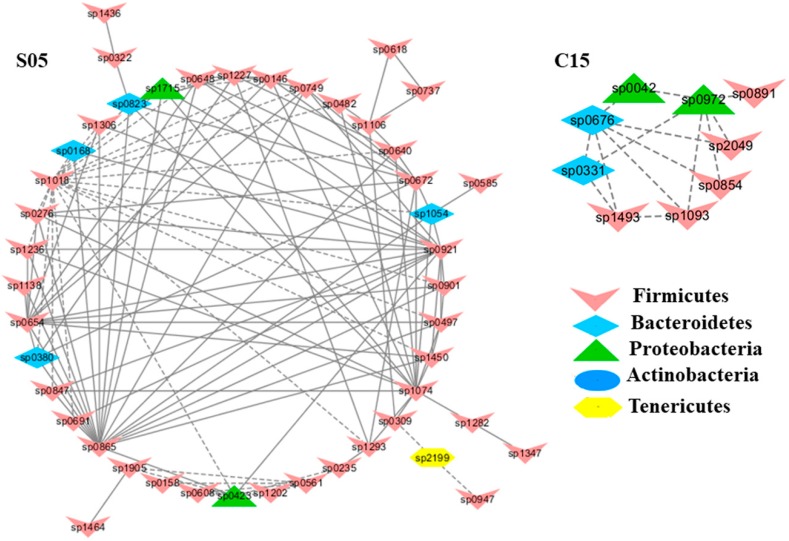
Modules C15 and S05 in the C and S network, respectively, showing a strong correlation with blood urea nitrogen levels (BUN). The detailed annotation of each OTU node can be found in Appendix A.

**Figure 4 ijms-20-02130-f004:**
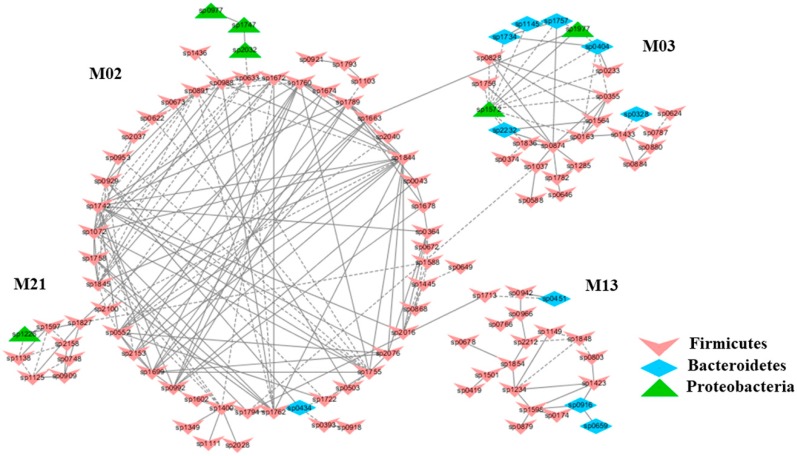
The modules in the mice subjected to exhaustive exercise stress (M) displaying a strong correlation with either blood urea nitrogen contents (BUN) (Module M03) or BUN to creatinine ratio (CR, Module M13) alone or both (Modules M02 and M21). The group N: healthy mice supplemented with PBS. The group M: mice subjected to exhaustive exercise stress supplemented with PBS. The group C: healthy mice supplemented with a daily dose of 150 mg/kg bodyweight of chondroitin sulfate disaccharide for 16 consecutive days. The group S: the stressed mice supplemented with a daily dose of 150 mg/kg bodyweight of CSD for 16 days. The detailed annotation of each OTU node can be found in Appendix A.

**Figure 5 ijms-20-02130-f005:**
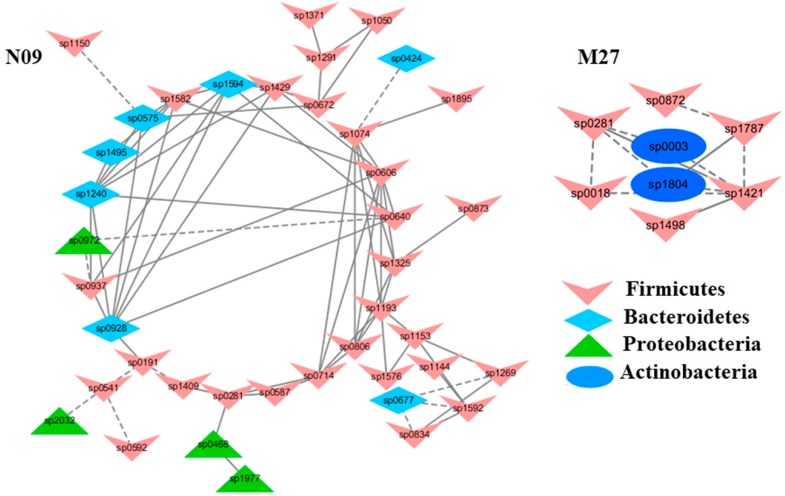
The modules with a strong correlation with kidney malondialdehyde concentrations (MDA), a lipid peroxidation marker. The interactions among different nodes (OTUs) within a module were shown: solid line: positive correlation; dashed line: negative correlation. The color of each node (OTU) indicated the phylum this OTU was assigned. Modules M27 and N09 were identified from the groups M and N, respectively. The group N: healthy mice supplemented with PBS; The group M: mice subjected to exhaustive exercise stress supplemented with PBS. The detailed annotation of each OTU node can be found in Appendix A.

**Figure 6 ijms-20-02130-f006:**
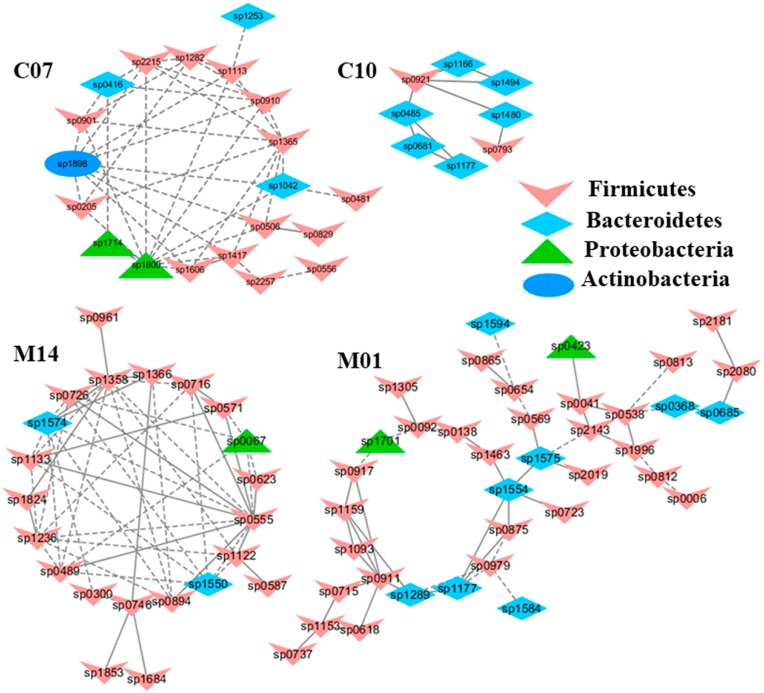
The modules displaying a strong correlation with kidney superoxide dismutase activities (SOD). The interactions among different nodes (OTUs) within a module were shown: solid line: positive correlation; dashed line: negative correlation. The color of each node (OTU) indicated the phylum this OTU was assigned. The modules C07 and C10, identified from the group C (healthy mice supplemented with a daily dose of 150 mg/kg bodyweight of chondroitin sulfate disaccharide for 16 consecutive days); showed a positive (C07) and negative (C10) correlation with SOD (*p* < 0.05), respectively. Similarly, Modules M01 and M14 displayed a positive (M01) and negative (M14) correlation with SOD in the group M, i.e., mice subjected to exhaustive exercise stress supplemented with PBS. The detailed annotation of each OTU node can be found in Appendix A.

**Table 1 ijms-20-02130-t001:** The correlations between the eigengene values of select modules and physiological traits. N: healthy mice supplemented with PBS; C: healthy mice supplemented with a daily dose of 150 mg/kg bodyweight of chondroitin sulfate disaccharide (CSD) for 16 consecutive days; M: mice subjected to exhaustive exercise stress supplemented with PBS; S: the stressed mice supplemented with a daily dose of 150 mg/kg bodyweight of CSD for 16 days. BUN: Blood urea nitrogen; CR: creatinine ratio; MDA: malondialdehydes; SOD: superoxide dismutase.

Module	Physiological Parameters	*r*	*p* Value	Module Members
Global Network/Group: N				
N01	BUN	0.68	0.040	34
N05	BUN	0.69	0.040	29
N14	CR	0.76	0.002	12
N09	MDA	0.67	0.050	40
Global Network/Group: M				
M02	BUN	0.90	0.002	54
M03	BUN	−0.89	0.003	29
M21	BUN	0.87	0.005	8
M02	CR	0.90	0.002	54
M21	CR	0.87	0.005	8
M13	CR	−0.65	0.080	20
M27	MDA	0.67	0.070	8
M01	SOD	0.71	0.050	38
M14	SOD	−0.72	0.050	22
Global Network/Group: S				
S05	BUN	−0.75	0.050	46
S02	CR	0.68	0.090	32
Global Network/Group: C				
C15	BUN	0.89	0.020	9
C01	CR	0.73	0.100	77
C03	CR	0.75	0.090	42
C04	CR	0.77	0.080	31
C07	SOD	0.76	0.080	20
C10	SOD	−0.94	0.006	8

**Table 2 ijms-20-02130-t002:** The partial Mantel test revealed the correlation between node connectivity of some taxa and the OTU significance of physiological traits in microbial co-occurrence networks. N: healthy mice supplemented with PBS; C: healthy mice supplemented with a daily dose of 150 mg/kg bodyweight of chondroitin sulfate disaccharide (CSD) for 16 consecutive days; M: mice subjected to exhaustive exercise stress supplemented with PBS; S: the stressed mice supplemented with a daily dose of 150 mg/kg bodyweight of CSD for 16 days. BUN: Blood urea nitrogen; CR: creatinine ratio; SOD: superoxide dismutase.

Treatment Group	Physiological Parameter	Taxon (Level)	*r* (Correlation Coefficient)	Significance (Probability)
N	BUN	Pseudomonodales (Order)	0.8144	0.0020
N	BUN	Porphyromonadaceae (Family)	0.9165	0.0250
N	BUN	Moraxellacease (Family)	0.7874	0.0040
C	CR	Pseudomonodales (Order)	0.6137	0.0250
C	BUN	Lactobacillacease (Family)	0.3149	0.0460
C	SOD	Lactobacillacease (Family)	0.5234	0.0039
M	SOD	Proteobacteria (Phylum)	0.5663	0.0250
S	CR	Pseudomonodales (Order)	0.6167	0.0333

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
