# Peer review of "Microbial Co-Occurrence Patterns and Keystone Species in the Gut Microbial Community of Mice in Response to Stress and Chondroitin Sulfate Disaccharide"

_ijms, 2019, doi:10.3390/ijms20092130_

Reviewer 1 Report

The authors have produced an interesting paper that posits a distinct microbiome mechanism for the actions of glucosamine and chondroitin sulfate which might be modestly effective yet alter physiological parameters consistent with vascular and renal dysfunction, however, it cannot be inferred as causative of disease. E.g. heavy exercise reduces GFR however there is no evidence that this as a result causes kidney disease. None the less I thought the authors made a plausible case for their study.

The method was sound and the analysis method was creative, while not being an expert on node modelling it seemed like a logical way to engage with the data yielded by this study. Likewise, the measurement methods were consistent with the theoretical investigation and well thought through.

The results section was well presented and clear

The discussion was logical and sequential, although I am not sure the issue of supposed renal toxicity was able to be resolved at any level based on the data. However, some interesting associations were observed which may explain the mechanisms for increased BUN. Overall the discussion was fine and this is certainly an area worthy of future investigation.

Both the animal and testing sections of the paper were fine.

Overall I thought this was a well written paper that did a good job connecting the dots with an inferential model. I think it can be published in its current form.

Author Response

Response to Reviewer 1 Comments

The authors have produced an interesting paper that posits a distinct microbiome mechanism for the actions of glucosamine and chondroitin sulfate which might be modestly effective yet alter physiological parameters consistent with vascular and renal dysfunction, however, it cannot be inferred as causative of disease. E.g. heavy exercise reduces GFR however there is no evidence that this as a result causes kidney disease. None the less I thought the authors made a plausible case for their study.

Response: We agree with the reviewer. No causative relationship can be established using network data alone. The results are nevertheless intriguing and should help us develop a testable hypothesis for future experiments. We modified the text accordingly.

The method was sound and the analysis method was creative, while not being an expert on node modelling it seemed like a logical way to engage with the data yielded by this study. Likewise, the measurement methods were consistent with the theoretical investigation and well thought through.

The results section was well presented and clear

Response: We appreciate the reviewer’s insightful comments and suggestions. 

The discussion was logical and sequential, although I am not sure the issue of supposed renal toxicity was able to be resolved at any level based on the data. However, some interesting associations were observed which may explain the mechanisms for increased BUN. Overall the discussion was fine and this is certainly an area worthy of future investigation.

Response: We agree and deleted the term renal toxicity from the discussion.

Both the animal and testing sections of the paper were fine.

Overall I thought this was a well written paper that did a good job connecting the dots with an inferential model. I think it can be published in its current form.

Response: Once again, we appreciated the reviewer’s suggestions and comments.

Reviewer 2 Report

The latest advancement in gut microbiota research had indicated that gut microbiota could influence human health. In the present manuscript, authors have provided valuable insights into the gut microbial interactions. Authors studied the microbial co-occurrence patterns using a random matrix theory-based approach in the gut microbiome of mice in response to chondroitin sulfate disaccharide in healthy and exercise-stressed conditions. Present study is executed well and adequately written and may be suitable for the publication after making following minor corrections.

Minor:

1. Section 2.1 of the result is hard to understand. Authors have started from the RNA-seq analysis. Discussing the objectives and experimental first would be helpful.

2. Figures 2 to 6 need professional improvements such as fonts could be smaller, and color combinations can be better.

3. Line 19: “key role for” should be “key role in”.

4. Line 44: “potential” should be “potentially”.

Author Response

Response to Reviewer 2 Comments

The latest advancement in gut microbiota research had indicated that gut microbiota could influence human health. In the present manuscript, authors have provided valuable insights into the gut microbial interactions. Authors studied the microbial co-occurrence patterns using a random matrix theory-based approach in the gut microbiome of mice in response to chondroitin sulfate disaccharide in healthy and exercise-stressed conditions. Present study is executed well and adequately written and may be suitable for the publication after making following minor corrections.

Response: We greatly appreciate the reviewer’s insightful comments.

Minor:

1. Section 2.1 of the result is hard to understand. Authors have started from the RNA-seq analysis. Discussing the objectives and experimental first would be helpful.

Response: We agree. Microbial interactions and co-occurrence patterns inferred using global network algorithms are not straightforward and difficult to understand. To overcome this, we included a substantial amount of data in Table S3. In addition, detailed procedures and parameters can be found in the Materials and Methods section after Discussion. However, we did not start the section with the RNAseq analysis. All results presented are closely related with 16S rRNA gene sequencing and network inference.

2. Figures 2 to 6 need professional improvements such as fonts could be smaller, and color combinations can be better.

Response: These figures are modified with a small font for annotation. Moreover, Table S3 was included for readers of interest to obtain detailed annotation or taxonomy info for each node in the modules.

3. Line 19: “key role for” should be “key role in”.

Response: Changed.

4. Line 44: “potential” should be “potentially”.

Response: Changed.